# Evaluation of Machine Learning Interatomic Potentials for Gold Nanoparticles—Transferability towards Bulk

**DOI:** 10.3390/nano13121832

**Published:** 2023-06-09

**Authors:** Marco Fronzi, Roger D. Amos, Rika Kobayashi

**Affiliations:** 1School of Chemical and Biomedical Engineering, University of Melbourne, Parkville, VIC 3010, Australia; 2School of Mathematical and Physical Sciences, University of Technology Sydney, Ultimo, NSW 2007, Australia; 3Supercomputer Facility, Australian National University, Canberra, ACT 2601, Australia; rika.kobayashi@anu.edu.au

**Keywords:** machine learning potentials, gold nanoparticles, molecular dynamics, structures, heat capacities

## Abstract

We analyse the efficacy of machine learning (ML) interatomic potentials (IP) in modelling gold (Au) nanoparticles. We have explored the transferability of these ML models to larger systems and established simulation times and size thresholds necessary for accurate interatomic potentials. To achieve this, we compared the energies and geometries of large Au nanoclusters using VASP and LAMMPS and gained better understanding of the number of VASP simulation timesteps required to generate ML-IPs that can reproduce the structural properties. We also investigated the minimum atomic size of the training set necessary to construct ML-IPs that accurately replicate the structural properties of large Au nanoclusters, using the LAMMPS-specific heat of the Au${}_{147}$ icosahedral as reference. Our findings suggest that minor adjustments to a potential developed for one system can render it suitable for other systems. These results provide further insight into the development of accurate interatomic potentials for modelling Au nanoparticles through machine learning techniques.

## 1. Introduction

From the point of view of chemical applications, it is only relatively recently that interest in gold nanoparticles has grown due to discoveries of their usefulness in several fields, including catalysis and biomedical applications [1,2,3,4]. Common methods utilised to investigate the properties of gold nanoparticles include quantum mechanical techniques, which provide high accuracy but are computationally demanding and often unfeasible. Molecular dynamics simulations rely on the quality of the underlying interatomic potentials, functions of the potential energy in terms of the atomic positions, and require costly ab initio calculations to obtain chemical accuracy. Machine Learning Interatomic Potentials (ML-IPs) directly target the potential energy surface through neural networks, thus avoiding costly calculations. In our previous work, we investigated their applicability to the properties of gold nanoparticles [5].

To benchmark against high-accuracy ab initio calculations, we limited ourselves to the 20-atom gold cluster and have discussed related works therein. However, in gold clusters containing up to 20 atoms, all of the gold atoms are located on the surface of the cluster, and it is only for clusters with more than 30 atoms that interior gold atoms become present. Significant internal structure starts to develop when there are around 50 atoms. This structural difference is a critical consideration for the physical and chemical properties of gold clusters, as surface and interior atoms may exhibit distinct electronic and chemical behaviours [6]. We have thus extended our previous study to larger clusters, progressively increasing cluster size towards bulk. Additionally, we have examined the question arising from our previous investigation of transferability and applicability, the ability of an ML-trained potential to produce meaningful results for systems other than that which it was trained on.

Previous density functional theory studies have explored the structural evolution and stability of gold clusters beyond 20 atoms, with density functional calculations by Nhat et al. revealing new stable forms in the range of 20 to 30 atoms [7]. Ouyang et al. [8] applied neural network potentials and the basin-hopping method to search for global minima of gold nanoclusters using Au${}_{58}$ as an example, as was also studied in an earlier work by Jensen and Jensen [9] using a capacitance–polarisability interaction model to reproduce the full polarisability tensors of medium-sized gold and silver clusters. Using geometrical constructs, Mori and Hegmann [10] proposed likely stable structures following a variety of classes of geometrical shapes to provide coordinates for clusters up to 1091 atoms. Additionally, optical excitation spectra of gold clusters of varying sizes have been investigated by Stener et al. [11], while Engel et al. [12] have shown that the structural and electronic properties of gold nanoparticles are influenced by the choice of support material, with the shape of the metal clusters dependent on the preferred isolated structure, the symmetry and distance of the preferred adsorption sites on the surface, and the relative strength of the metal–metal interactions to the metal surface interactions.

As the size of a particle increases, we reach a limit where the surface atoms and their interactions become negligible when compared to the bulk-like atoms that are chemically inert [13]. To simulate this approximation, periodic boundary conditions can be applied to a relatively small conventional or primitive cell, which enables the prediction of structural properties rather than functional properties. In contrast to crystalline nanoparticles, amorphous Au nanoparticles lack a well-defined lattice structure and have a more complex electronic structure due to the disordered atomic arrangement. The disordered surface results in a larger number of low-coordination sites that are highly active catalytic sites, which can be beneficial for application in catalysis [14].

The optical properties of bulk gold were first studied by Christensen et al. [15], who calculated the energy band structure using the relativistic augmented-plane-wave method. Their calculations revealed that the shifts and splittings due to relativistic effects were of the same order of magnitude as the gap.

## 2. Computational Details

In our previous investigation [5], we performed extensive MD calculations on the lowest energy structure of the Au${}_{20}$ nanocluster (Figure 1) and two isomers with the Vienna Ab Initio Simulation Package (VASP) [16,17], to use as training set to build an interatomic potential. The DeePMD package [18,19] was then used to parameterise the potential and calculate properties with the Large-scale Atomic/Molecular Massively Parallel Simulator (LAMMPS) [20] package. We found good agreement between the two approaches as discussed in that work.

We have continued this methodology for the current investigation, adapting for size. The cubic supercell, with a lattice parameter of 30 Å, encompassed all finite clusters, with up to 147 atoms, of the structures examined, whereas a (2 × 2 × 2) expansion of the conventional Au cell has been used for the bulk calculations. The Perdew–Burke–Ernzerhof (PBE) generalised gradient approximation (GGA) was used for the exchange–correlation functional [21]. VASP core–valence interaction was described using projector augmented-wave (PAW) potentials, with periodic plane waves used to represent valence electrons with a cutoff energy of 520 eV. The energy and forces tolerance for structural relaxation were 10${}^{-6}$ eV and 10${}^{-6}$ eVÅ${}^{-1}$, respectively. Equations of motion were integrated using the velocity Verlet algorithm with a time step of 2.5 fs. Temperature stabilisation of the system was achieved through the use of a Nosé–Hoover thermostat, with a temperature-damping parameter of 10 time steps. This deterministic method relies on the extended system idea, introducing an additional degree of freedom *s* to represent the thermal reservoir, ensuring ergodicity within the canonical ensemble was used to calculate properties.

Calculations for the amorphous structures were obtained from VASP and LAMMPS calculations by first running at elevated temperature, and then quenching the resulting structure to 0 K through a geometry relaxation.

## 3. Results and Discussion

With the question of transferability in mind, to explore whether the potential developed with the Au${}_{20}$ data is able to describe other small gold nanoclusters, we used this potential to look at the structures of Au${}_{22}$, Au${}_{24}$, Au${}_{26}$ Au${}_{28}$, and Au${}_{30}$. Optimised structures obtained for Au${}_{30}$ are given in Figure 2. These systems, and others, have already been examined by Nhat et al. [7]. Generally, the predicted structures were topologically correct in the sense the clusters had the correct overall shape with the atoms in approximately the correct position. However, it was noticed that the bond lengths were systematically predicted to be too long. For example, Figure 2a displays the structure predicted using the potential derived from the Au${}_{20}$ data and many of the atoms are sufficiently far apart that they do not qualify as having bonds. A geometry optimisation with VASP produces a more compact structure, as in Figure 2b, with shorter bonds. Accordingly, the potential was refined by including extra data obtained from geometry optimisations on Au${}_{28}$ and Au${}_{30}$. This is a comparatively small amount of data, with only 80 energy and force evaluations for Au${}_{28}$ and 71 for Au${}_{30}$. The data from Au${}_{28}$ were added to the training set, and the Au${}_{30}$ data to the validation set, and the training of the potential with DeePMD was repeated. With this revised potential, the structure of Au${}_{30}$ predicted using LAMMPS is shown in Figure 2c, and can be seen to be much closer to the VASP result. Note that the data for Au${}_{30}$ were not used for training, so this result is a genuine prediction. The results for Au${}_{28}$ are also much improved. The coordinates of the final VASP- and LAMMPS-optimised structures are given in Appendix A.

The LAMMPS structure of Au${}_{30}$ is not in perfect agreement with that from VASP, but it is sufficiently improved by just the addition of a few extra data points, such that it was worth seeing whether this process could be extended. Therefore, we systematically expanded the size of the clusters considered. These small clusters examined so far are all of cage-like topology, but there is nothing (except for one atom in the case of Au${}_{30}$) inside the cage; i.e., for these examples, all the atoms are on the surface of the cluster. This can be seen if one looks at the structures given by Nhat et al. [7]. As mentioned in the introduction, significant internal structure in nanoclusters starts to develop when there are around 50 atoms (see for example Ouyang et al. [8]), and so we next considered the isomers of Au${}_{55}$ and Au${}_{58}$. There are suggested structures [10] for a large number of potential nanoclusters, including three for Au${}_{55}$, based on geometric arguments rather than calculations. We used three proposed Au${}_{55}$ structures, described as Cube, Icosahedron and Decahedron. Coordinates for these are given in the supplementary data of Ref. [10]. There are also published calculations on Au${}_{58}$ in Refs. [8,9]. As with Au${}_{30}$, it was found necessary to make an adjustment to the potential, using additional geometry optimisations with VASP. The calculations on Au${}_{55}$ (190 energy and force evaluations across all 3 isomers) were added to the training set, and those on Au${}_{58}$ (71 evaluations) to the validation data, and the training with DeePMD was repeated. The VASP structures for Au${}_{55}$ and the resulting equivalent LAMMPS results using the updated potential are shown in Figure 3.

The predicted structures of the Au${}_{55}$ isomers with VASP and LAMMPS are in close agreement, sufficiently so that it is difficult to tell the two set of results apart from just the images (the images in the figure are those from VASP). The full coordinates from both sets of calculations are given in Appendix A . The closeness of the two structures can be seen by comparing the bond lengths. For the icosahedral structure, this is shown in Figure 4.

There are 3 (or possibly 4) groups of bond lengths, and the values obtained from VASP and LAMMPS are in close agreement. The root mean square difference (RMSD) between the 2 set of values is 0.006 Å. The equivalent results for the cube and decahedron versions of Au${}_{55}$ show RMSD values of 0.010 Å in both cases. The equivalent plots are in Appendix A.

The predicted geometries for Au${}_{58}$ from VASP and LAMMPS, using the same potential in LAMMPS as for Au${}_{55}$, are in Figure 5 and Appendix A. Again, the structures agree closely visually. Note that Au${}_{58}$ is not included in the training data, so this is a prediction.

However, the bonding in Au${}_{58}$ is very different to that in the Au${}_{55}$ isomers. The latter have much symmetry and only a few unique bond lengths, as seen in Figure 4, but in Au${}_{58}$ the bonds are all different; i.e., it is an amorphous structure, as in Figure 6. The VASP and LAMMPS structures have an RMSD in the bond lengths of 0.038 Å.

The clusters Au${}_{55}$ and Au${}_{58}$ are large enough to have some internal structure, and having adjusted the potential to describe these, it was worth checking whether it could now provide the structures of still larger clusters. By expanding to about 150 atoms, the internal structure becomes more complex. Mori and Hegmann [10] have approximate structures for Au${}_{147}$ based on geometric arguments rather than actual calculations, and there exist other calculations on Au${}_{147}$; e.g., Ref. [22]. It is found that geometry relaxations on these approximate structures [10] using VASP and LAMMPS agree, as in Figure 7, without any further re-parameterisation, i.e, the potential which described systems up to Au${}_{58}$ also works for Au${}_{147}$.

Like the structures for Au${}_{55}$, it is not possible to distinguish the VASP and LAMMPS structures just from the images (full coordinates of all isomers are in Appendix A) of the Appendix A. The bond lengths show the pattern seen in Au${}_{55}$, with comparatively few unique values, and close agreement between VASP and LAMMPS. The result for the icosahedral isomer is shown in Figure 8, with the equivalent graphs for the other two isomers in Appendix A. The RMSD for the bond lengths is 0.015, 0.018 and 0.015 Å for the icosahedral, decahedral and cubic isomers, respectively.

Reproducing the VASP results for Au${}_{147}$ with a potential parameterised on smaller systems is an interesting and promising result. However, it should be noticed most structures mentioned so far are symmetric, and there are suggestions that these particles are actually amorphous, i.e., irregular, non-crystalline structures( e.g., Ref. [22]) so there may be many low-lying minima on the Au${}_{147}$ surfaces which could be examined.

Figure 9 shows two amorphous structures of Au${}_{147}$ (coordinates in Appendix A). There are probably many such structures, so the images may correspond to different local minima. What is significant, however, is that in both cases the energy is lower than that of the most stable symmetric structure, the isocahedral configuration; see Figure 7b. Tarrat et al. [22], using DFT calculations, found many such structures, so it is not surprising that our VASP calculations could also find an amorphous example, but it is interesting that our ML-IP can reproduce this, even though it was not parameterised on this system.

Another caveat is that although the structures are reproduced correctly, the energy differences between various isomers of Au${}_{55}$ and Au${}_{147}$ from LAMMPS may not be reliable, though they are at least in the same order as the VASP calculations.

Regarding the question of time and size thresholds, in Ref. [5] we used many data points to determine the ML-IAP of Au${}_{20}$ to ensure we were benchmarking at a high accuracy. This made us wonder whether similar accuracy could be achieved with less data. Accordingly, we tested the accuracy of the ML-IPs generated from six different VASP MD timescales by calculating the specific heat of a structure of Au${}_{20}$. We chose the global minimum (c.f. Figure 1) due to its stable energy configuration with respect to the other isomers considered in our previous work. The specific heat was calculated using the energy fluctuation model, as reported in our previous work. The C${}_{v}$ converge when the ML-IPs were generated using more than 5300 simulation timesteps; see Table 1. This is in line with the ML optimisation tests that indicate discrepancies in the forces between the training and validation for simulation lengths below 5300 timesteps. Figure 10 shows the convergence of the training process showing the RMSE of the training and validation sets for each set of data. In particular, it should be noted that the energy converges with much fewer points than the force, which is of relevance as the specific heat calculation depends on the forces.

The scalability of the ML-IP was investigated using an Au${}_{147}$ icosahedron as reference. Four ML-IPs were generated by considering VASP MD performed on either one structure or the combination of two structures. Specifically, the ML-IPs are labelled IP-Au20, IP-Au20+bulk, IP-Au147-ico and IP-Au147-amorph, where the ML-IPs were generated from Au${}_{20}$, Au${}_{20}$ and bulk Au, icosahedral Au${}_{147}$ and amorphous Au${}_{147}$ VASP MD simulations, respectively. The aim was to determine the smallest number of atoms, contained in a cell used in VASP MD simulation, required for constructing a “universal” ML-IP. To check the accuracy of ML-IPs, LAMMPS molecular dynamics simulations were performed, and the specific heats of the NPs were calculated from energy fluctuations.

To assess the accuracy of the specific heat calculated using the four ML-IPs in LAMMPS simulations, the Au147-ico specific heat was considered the reference value of 20.01 J/K/mol (see Table 2). The results show that using IP-Au20 resulted in an error of 19.65% with respect to the one calculated using IP-Au147-ico. However, using IP-Au20+bulk significantly increased the accuracy, reducing the relative error to 3.76%. Additionally, using IP-Au147-amorph resulted in an error of 2.34%. The results suggest that a good ML-IP can be generated by considering a training set that includes information on the surface and bulk, without the need for additional information. Indeed, the ML-IP generated by only considering VASP bulk and Au${}_{20}$ simulations helped to recover the error caused by the large surface-to-volume ratio of a small nanocluster.

## 4. Conclusions

Our preliminary investigation of the applicability of ML-IPs to the properties of gold nanoparticles [5] highlighted many areas for further investigation, largely related to accuracy and reducing time-to-solution of our calculations. Of main interest was the transferability of the potentials we had already created to other systems. In order to explore this, we extended our calculations to larger nanoclusters, firstly looking at Au${}_{28}$ and Au${}_{30}$. We found that the original potential developed for Au${}_{28}$ was not sufficient but all that was needed was a few extra training points obtained from geometry optimisations. Subsequently, we extended the study to multiple structures of Au${}_{55}$ and Au${}_{58}$, which is the size at which the clusters acquire an internal structure. Again, we found that a few extra geometry calculations (on Au${}_{55}$) were sufficient. This updating process resembles the idea of “delta learning” as illustrated in other papers, whereby a potential is used until it exceeds the region for which it has been trained, and then it is updated; see, e.g., Ref. [23]. Finally, this last potential could be used for Au${}_{147}$ without further modification, which is a useful indication that potentials developed on one system can be applied to another.

## Figures and Tables

**Figure 1 nanomaterials-13-01832-f001:**
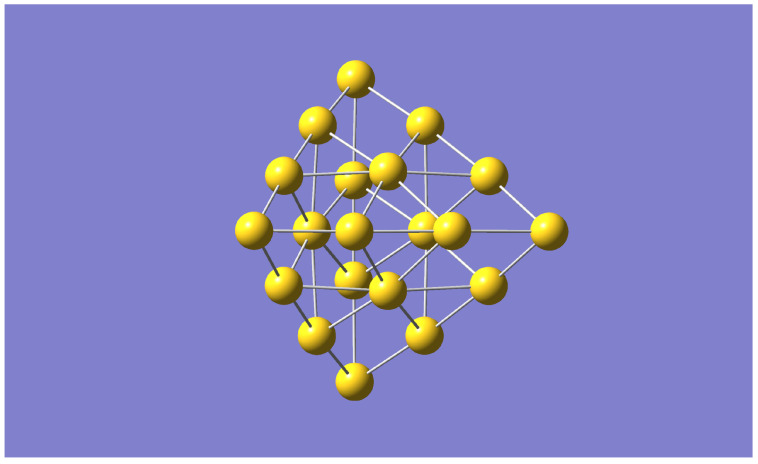
The Gold Au${}_{20}$ structure.

**Figure 2 nanomaterials-13-01832-f002:**

Structures of Au${}_{30}$ produced (**a**) using the potential trained on Au${}_{20}$, (**b**) using VASP and (**c**) using a modified ML potential.

**Figure 3 nanomaterials-13-01832-f003:**
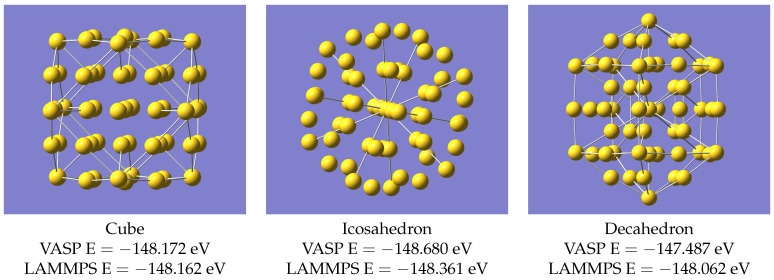
Structures of Au${}_{55}$ from VASP.

**Figure 4 nanomaterials-13-01832-f004:**
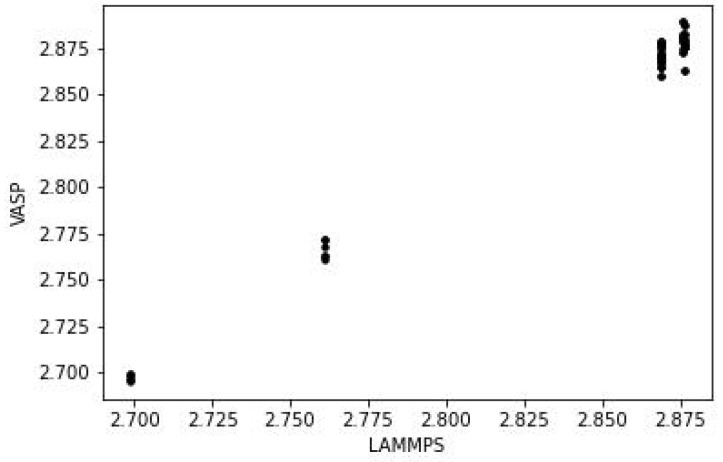
Comparison of bond lengths (in Å) of icosahedral Au${}_{55}$ from VASP and LAMMPS.

**Figure 5 nanomaterials-13-01832-f005:**
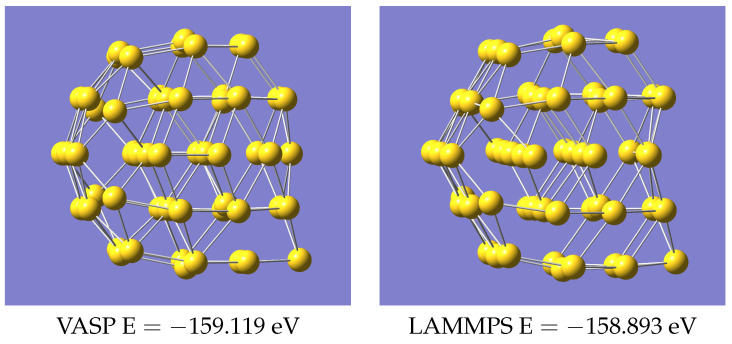
Structures of Au${}_{58}$ using VASP and LAMMPS.

**Figure 6 nanomaterials-13-01832-f006:**
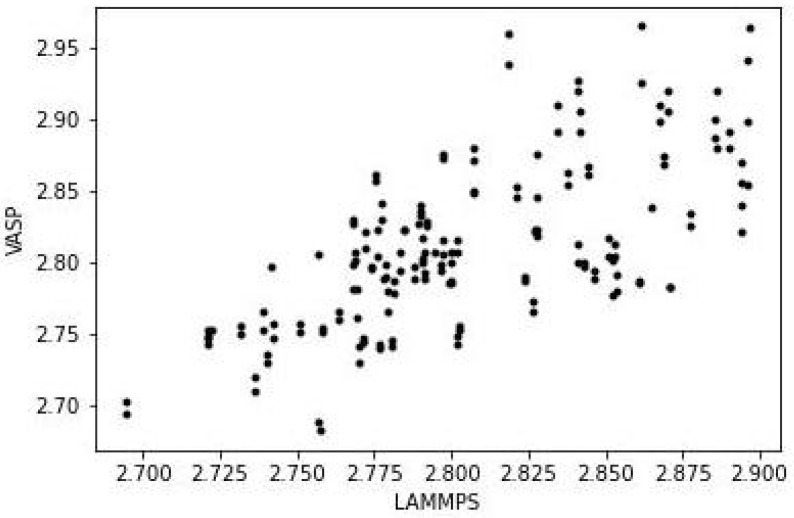
Comparison of bond lengths (in Å) of Au${}_{58}$ from VASP and LAMMPS.

**Figure 7 nanomaterials-13-01832-f007:**
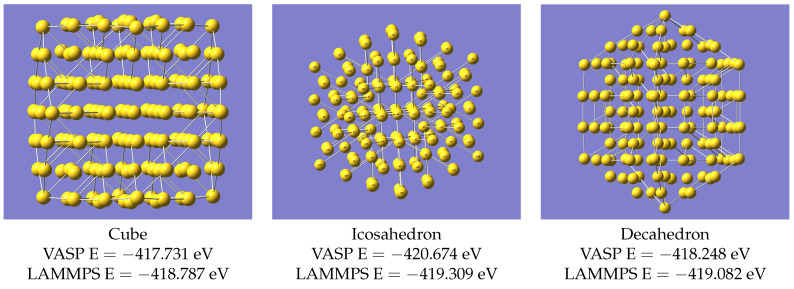
Structures of Au${}_{147}$ computed with VASP.

**Figure 8 nanomaterials-13-01832-f008:**
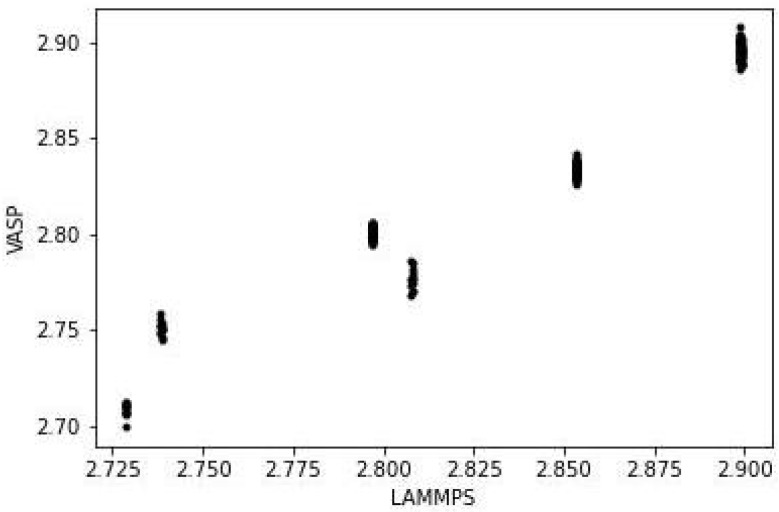
Comparison of bond lengths (in Å) of icosahedral Au${}_{147}$ from VASP and LAMMPS.

**Figure 9 nanomaterials-13-01832-f009:**
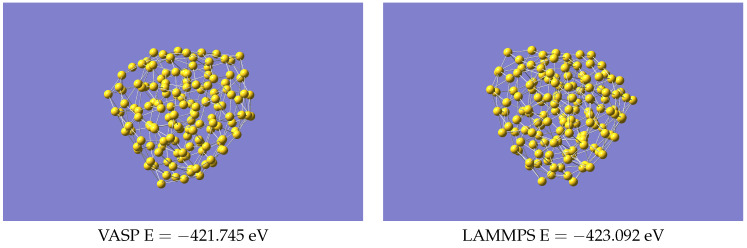
Amorphous Structures of Au${}_{147}$ using VASP and LAMMPS.

**Figure 10 nanomaterials-13-01832-f010:**
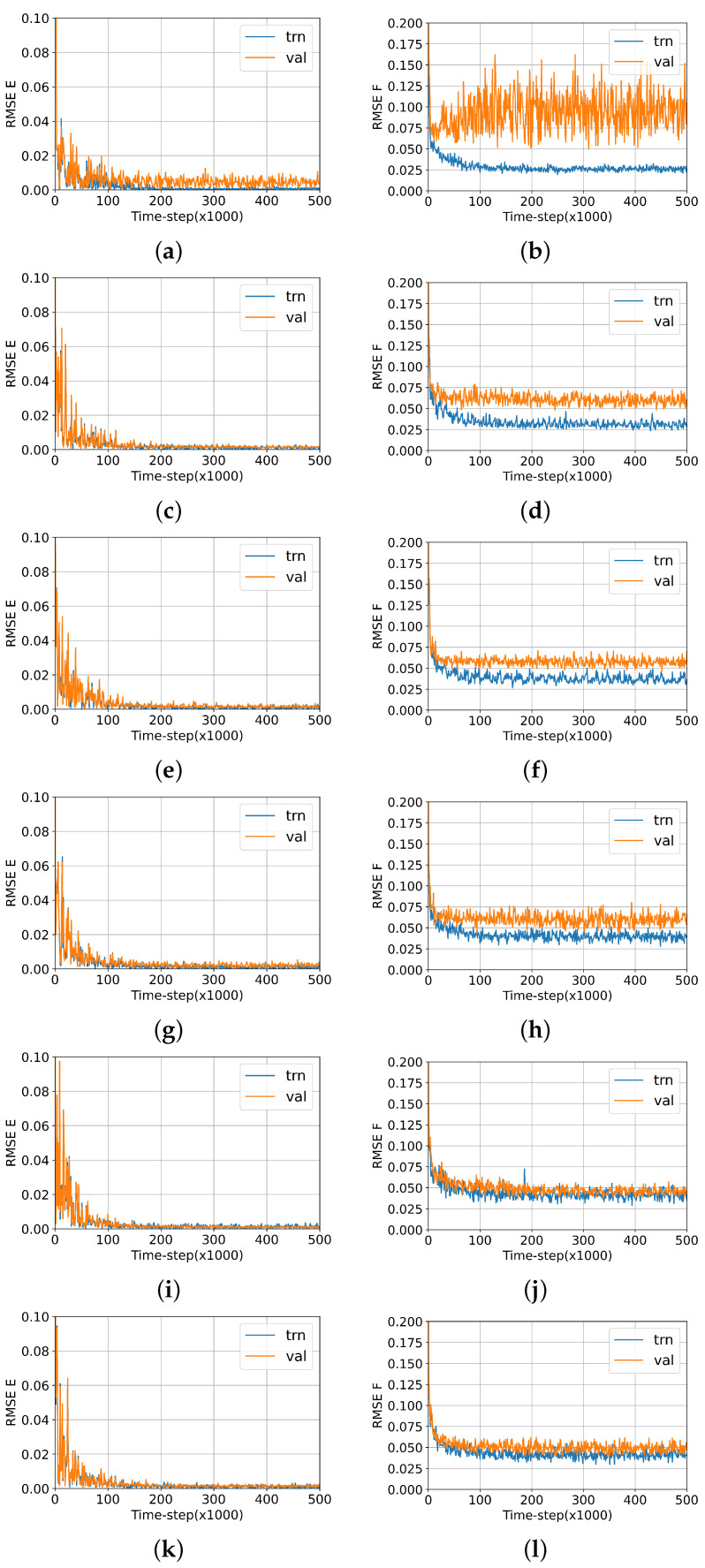
DeepMD Energy (left panels) and Force (right panels) optimisation convergence test, performed using six different VASP simulation lengths on the Au${}_{20}$ global minimum: 700 (**a**,**b**), 1300 (**c**,**d**), 2700 (**e**,**f**), 4000 (**g**,**h**), 5300 (**i**,**j**) and 7000 (**k**,**l**).

**Table 1 nanomaterials-13-01832-t001:** Specific heat of Au${}_{20}$ global minimum calculated by energy fluctuation of LAMMPS simulations. Five different ML-IPs were generated using different VASP MD simulations timelengths. The Timesteps column indicates the length of the ab initio simulations used to generate the ML-IPs.

Specific Heat	
Timesteps	C${}_{v}$(J/K/mol)
700	16.67
1300	17.79
2700	17.17
4000	16.98
5300	19.05
7000	18.95

**Table 2 nanomaterials-13-01832-t002:** Specific heat relative errors calculated using four different ML-IPs in LAMMPS MD simulations. The errors are relative to the Au147-ico MLIP C${}_{v}$ = 20.01 J/K/mol.

Specific Heat	
ML-IP	Relative Error (%)
IP-Au20	19.65
IP-Au20+bulk	3.76
IP-Au147-amorph	2.34
IP-Au147-ico	0.00

## Data Availability

Coordinates of all structures described in this study are provided in the Appendix A.

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
