# Peer review of "Evaluation of Machine Learning Interatomic Potentials for Gold Nanoparticles—Transferability towards Bulk"

_nanomaterials, 2023, doi:10.3390/nano13121832_

Round 1

Reviewer 1 Report

The paper is dedicated to efficacy of machine learning (ML) interatomic potentials (IP) in modeling gold (Au) nanoparticles. Authors investigated the minimum atomic size of the training set necessary to construct ML-IPs that accurately replicate the structural properties of large Au nanocluster. Obtained simulation results provide further insight into the development of accurate interatomic potentials for modeling Au nanoparticles through machine learning techniques.

Authors made an interesting research for modeling gold nanoparticles. Although there is no information about correlation of theoretical results and experimental data from literature. Authors should moreover be more concentrated on the practical application of their research to publish this article in “Nanomaterials” journal.

The following claims should be also addressed by authors before publishing:  

1.  Line 25 ”However, in gold clusters containing up to 20 atoms, all of the gold atoms are located on the surface of the cluster, and it is only for clusters with more than 30 atoms that interior gold atoms become present. Significant internal structure starts to develop when there are around 50 atoms.” This statement needs a proof reference. Does it come from
6. Zhou, M.; Du, X.; Wang, H.; Jin, R. The Critical Number of Gold Atoms for a Metallic State Nanocluster: Resolving a Decades-Long Question. ACS Nano (2021), 15, 13980-13992?

2. Line 54 “In contrast to crystalline nanoparticles, amorphous Au nanoparticles lack a well-defined lattice structure and have a more complex electronic structure due to the disordered atomic arrangement.” In terms of nanoparticle, what is understood under amorphous state should be additionally discussed.

3.  Are all atoms on Figure 1 situated on the surface of the cluster? Maybe it could be visualized somehow else? Using i.e. cross-section

4. There should be a clear comparison between presented simulation results and those already published by the authors and other groups to demonstrate the article’s novelty, i.e. using a table.

5. What are the dimensions for figs 4,6,8? Is it Angstroms? It should be marked

6. Line 190 “The scalability of the ML-IP was investigated using an Au147 icosahedron as reference”. Why?

7. There should be a misprint in line 194 “Au14taxol. f ull.newout2.48pe7”

8.  Line 224 “we haven’t a conclusion for the second part? What does it mean?

9. For experimental results on island films growth author could check: https://doi.org/10.1038/s41598-019-48508-3, https://doi.org/10.1002/pssc.200982480

10.  I was not able to find Supplementary Material. Could the authors send them to [email protected]

The paper is dedicated to efficacy of machine learning (ML) interatomic potentials (IP) in modeling gold (Au) nanoparticles. Authors investigated the minimum atomic size of the training set necessary to construct ML-IPs that accurately replicate the structural properties of large Au nanocluster. Obtained simulation results provide further insight into the development of accurate interatomic potentials for modeling Au nanoparticles through machine learning techniques.

Authors made an interesting research for modeling gold nanoparticles. Although there is no information about correlation of theoretical results and experimental data from literature. Authors should moreover be more concentrated on the practical application of their research to publish this article in “Nanomaterials” journal.

The following claims should be also addressed by authors before publishing:  

1. Line 25 ”However, in gold clusters containing up to 20 atoms, all of the gold atoms are located on the surface of the cluster, and it is only for clusters with more than 30 atoms that interior gold atoms become present. Significant internal structure starts to develop when there are around 50 atoms.” This statement needs a proof reference. Does it come from 6. Zhou, M.; Du, X.; Wang, H.; Jin, R. The Critical Number of Gold Atoms for a Metallic State Nanocluster: Resolving a Decades-Long Question. ACS Nano (2021), 15, 13980-13992?

2. Line 54 “In contrast to crystalline nanoparticles, amorphous Au nanoparticles lack a well-defined lattice structure and have a more complex electronic structure due to the disordered atomic arrangement.” In term of nanoparticle it should be additionally discussed what is understood under amorphous state.

3. Are all atoms on Figure 1 situated on the surface of the cluster? Maybe it could be visualized somehow else? Using i.e. cross section

4. There should be a clear comparison between presented simulation results and already published by the authors and other groups to demonstrate article’s novelty, i.e. using a table.

5. What is the dimensions for figs. 4,6,8? Is it Angstroms? It should be marked

6.  Line 190 “The scalability of the ML-IP was investigated using an Au147 icosahedron as reference”. Why?

7.  There should be a misprint in line 194 “Au14taxol. f ull.newout2.48pe7”

8.  Line 224 “we haven’t a conclusion for the second part? What does it mean?

9. For experimental results on island films growth author could check: https://doi.org/10.1038/s41598-019-48508-3, https://doi.org/10.1002/pssc.200982480

10.  I was not able to find Supplementary Material. 

Author Response

Reviewer 1:
1. Line 25 ”However, in gold clusters containing up to 20 atoms, all of the gold atoms are located on the surface of the cluster, and it is only for clusters with more than 30 atoms that interior gold atoms become present. Significant internal structure starts to develop when there are around 50 atoms.” This statement needs a proof reference. Does it come from  6. Zhou, M.; Du, X.; Wang, H.; Jin, R. The Critical Number of Gold Atoms for a Metallic State Nanocluster: Resolving a Decades-Long Question. ACS Nano (2021), 15, 13980-13992? The simplest way to see the structures of the smaller particles is to visualise the Supplementary Data in Nhat et al. which contain all the structures from Au 20 Au 30 and show there is no atom inside the cage. We have added a comment pointing to this in the discussion section.

2. Line 54 “In contrast to crystalline nanoparticles, amorphous Au nanoparticles lack a well defined lattice structure and have a more complex electronic structure due to the disordered atomic arrangement.” In terms of nanoparticle, what is understood under amorphous state should be additionally discussed. Our paper explores an example of amorphous configuration of Au, obtained by a randomized cleavage of a cubic Au structure. After a short AIMD, we observed the structure's relaxation into an amorphous form. Subsequent geometry optimization showed that this amorphous
structure had a lower energy compared to crystalline forms. Typically, a lower total energy suggests an enhanced thermodynamic stability, as a higher thermal energy is needed to break stronger chemical bonds. This could be relevant when considering the complex trend in melting points exhibited by Au structures. However, no conclusion can be obtained from the analysis presented here, because this would require a much more extensive analysis, which however deviates from the scope of this paper.

3. Are all atoms on Figure 1 situated on the surface of the cluster? Maybe it could be visualized somehow else? Using i.e. cross section Unfortunately, the only way to see this is to use a molecular viewer and rotating them. Readers should be able to download free software to do this with the coordinates provided in the Supplementary Material.

4. There should be a clear comparison between presented simulation results and already published by the authors and other groups to demonstrate article’s novelty, i.e. using a table. I am not sure how we can do this. Our results should be the same as obtained by other people or better through the use of high-accuracy ab initio methods which are trusted to have high accuracy. The novelty is that through the use of ML-IP we can get these results faster, from hundreds of hours to minutes. As no-one really provides timings for these calculations it will be difficult to provide a meaningful table.

5. What is the dimensions for figs. 4,6,8? Is it Angstroms? It should be marked Yes, Angstroms – now added into captions.

6. Line 190 “The scalability of the ML-IP was investigated using an Au147 icosahedron as reference”. Why? The 147 structures are the largest considered in the paper, in terms of number of atoms, and the primary target of the paper is to maintain accuracy when ML-IP are used for large NP. Our choice to reference quantities with respect to the Au147 structure was made with an aim to underscore this point.
7. There should be a misprint in line 194 “Au14taxol. f ull.newout2.48pe7” Fixed – rogue cut and paste. Thank you for pointing this out.

8. Line 224 “we haven’t a conclusion for the second part? What does it mean? This line was part of the penultimate draft which was submitted by accident. It was not in the final version.

9. For experimental results on island films growth author could check: https://doi.org/10.1038/s41598-019-48508-3, https://doi.org/10.1002/pssc.200982480 We thank the referee for bringing these papers to our attention but they are not sufficiently close to the subject area of our paper to be included.

10. I was not able to find Supplementary Material.  We think this is a consequence of the MDPI LaTeX template which provides the text before the link is made. The Supplementary Material cannot be accessed from the link in the manuscript but should be accessible from the reviewer page. Nevertheless, we have emailed the Material.

Reviewer 2 Report

The authors analyse the efficacy of machine learning (ML) interatomic potentials (IP) in modeling gold nanoparticles. They explored transferability of these ML models to larger systems.

This paper follows a previous paper from the same authors showing applicability of ML to clusters of gold nanoparticles. Although I believe the authors discussed ML application to IP in their previous work, I strongly suggest that the author add a section where it is reviewed/described how ML is employed because this is not a familiar well-established topic.

no comment

Author Response

Reviewer 2:
This paper follows a previous paper from the same authors showing applicability of ML to clusters of gold nanoparticles. Although I believe the authors discussed ML application to IP in their previous work, I strongly suggest that the author add a section where it is reviewed/described how ML is employed because this is not a familiar well-established topic. This is a fair point. We have added an introductory explanation as follows so this paper can
be read standalone.

“Molecular dynamics simulations rely on the quality of the underlying interatomic potentials, functions of the potential energy in terms of the atomic positions, and require costly ab initio calculations to obtain chemical accuracy. Machine Learning Interatomic Potentials (ML-IP) directly target the potential energy surface through neural networks thus avoiding the costly calculations”

Reviewer 3 Report

Fronzi et al. continued the study on improving the interaction potential for Au clusters use a machine learning approach. They tested the transferability of the improved potential for clusters containing up to 147 Au atoms. The present potential is considered to work for Au clusters of various sizes. The results might be useful for people working in the related fields. The approaches were described in detail. The manuscript is written properly. The text is in this Journal. Therefore, I’d like to suggest acceptance of this manuscript for publication in Nanomaterials after some improvements.

1. The introduction can be improved in a more logical way.

2. It might be better to use systematically the same abbreviation, e.g. the unit of bonds (Å) instead of Ångstrom.

3. At the bottom of Conclusions, the sentence in italics seems strange (line 224).

4. Some typos can be corrected, e.g. in Ref. 23 (line 285).

The English is OK, except for some typos.

Author Response

Reviewer 3:
1. The introduction can be improved in a more logical way. We have attempted to improve the introduction as requested by Referee 2 and after discovering a confusing typo reformatted the paragraphs to emphasise the logical progression of related work on nanoparticles increasing in size from small clusters through amorphous to
bulk.

2. It might be better to use systematically the same abbreviation, e.g. the unit of bonds (Å) instead of Ångstrom.
We have made these all consistent now.

3. At the bottom of Conclusions, the sentence in italics seems strange (line 224). This line was part of the penultimate draft which was submitted by accident. It is not in the final version.

4. Some typos can be corrected, e.g. in Ref. 23 (line 285). Fixed

Round 2

Reviewer 1 Report

Unfortunately, updated version still does not reach Nanomaterials journal level